# Usefulness of Mesenchymal Cell Lines for Bone and Cartilage Regeneration Research

**DOI:** 10.3390/ijms20246286

**Published:** 2019-12-13

**Authors:** M. Piñeiro-Ramil, C. Sanjurjo-Rodríguez, R. Castro-Viñuelas, S. Rodríguez-Fernández, I.M. Fuentes-Boquete, F.J. Blanco, S.M. Díaz-Prado

**Affiliations:** 1Grupo de Investigación en Terapia Celular e Medicina Rexenerativa, Departamento de Fisioterapia, Medicina e Ciencias Biomédicas, Facultade de Ciencias da Saúde, Universidade da Coruña (UDC), Campus de A Coruña, 15006 A Coruña, Spain; clara.sanjurjo@udc.es (C.S.-R.); rocio.castro@udc.es (R.C.-V.); s.rodriguezf@udc.es (S.R.-F.);; 2Grupo de Investigación en Terapia Celular e Medicina Rexenerativa, Instituto de Investigación Biomédica de A Coruña (INIBIC), Complexo Hospitalario Universitario A Coruña (CHUAC), Servizo Galego de Saúde (SERGAS), Universidade da Coruña (UDC), 15006 A Coruña, Spain; 3Grupo de Investigación en Terapia Celular e Medicina Rexenerativa, Centro de Investigacións Científicas Avanzadas (CICA), Agrupación Estratéxica entre o CICA e o Instituto de Investigación Biomédica de A Coruña (INIBIC), Universidade da Coruña (UDC), 15071 A Coruña, Spain; 4Centro de Investigación Biomédica en Red (CIBER) de Bioingeniería, Biomateriales y Nanomedicina (CIBER-BBN), 28029 Madrid, Spain; fblagar@sergas.es; 5Grupo de Investigación en Reumatología (GIR), Instituto de Investigación Biomédica de A Coruña (INIBIC), Complexo Hospitalario Universitario A Coruña (CHUAC), Servizo Galego de Saúde (SERGAS), 15006 A Coruña, Spain

**Keywords:** cartilage and bone repair, immortalization, mesenchymal stromal cells, cell therapy, tissue engineering

## Abstract

The unavailability of sufficient numbers of human primary cells is a major roadblock for in vitro repair of bone and/or cartilage, and for performing disease modelling experiments. Immortalized mesenchymal stromal cells (iMSCs) may be employed as a research tool for avoiding these problems. The purpose of this review was to revise the available literature on the characteristics of the iMSC lines, paying special attention to the maintenance of the phenotype of the primary cells from which they were derived, and whether they are effectively useful for in vitro disease modeling and cell therapy purposes. This review was performed by searching on Web of Science, Scopus, and PubMed databases from 1 January 2015 to 30 September 2019. The keywords used were ALL = (mesenchymal AND (“cell line” OR immortal*) AND (cartilage OR chondrogenesis OR bone OR osteogenesis) AND human). Only original research studies in which a human iMSC line was employed for osteogenesis or chondrogenesis experiments were included. After describing the success of the immortalization protocol, we focused on the iMSCs maintenance of the parental phenotype and multipotency. According to the literature revised, it seems that the maintenance of these characteristics is not guaranteed by immortalization, and that careful selection and validation of clones with particular characteristics is necessary for taking advantage of the full potential of iMSC to be employed in bone and cartilage-related research.

## 1. Introduction

Therapeutic options capable of restoring the physiological properties of bone and cartilage are still lacking [1,2]. Due to the increase in life expectancy of the population, the incidence of musculoskeletal disorders, such as fractures, osteoporosis, maxillofacial pathologies, and rheumatic diseases such as osteoarthritis, is rising [3,4]. In this context, tissue engineering has emerged as a potential alternative treatment that could provide biological tissue substitutes for replacing the damaged ones, using scaffolds and cells. However, although many efforts have been made, very few tissue engineering techniques have been translated into clinical practice, and the ideal scaffold for engineering bone and cartilage substitutes has not yet been developed [1,4,5,6].

Tissue engineering techniques, combining scaffolds and cells, must undergo in vitro testing before translation into clinic or preclinical models. Human mesenchymal stromal cells (MSCs) are often employed in bone and cartilage tissue engineering approaches because of their proliferation and multidifferentiation abilities [7,8,9]. However, much more research is still needed to optimize isolation, expansion, differentiation, and preconditioning of MSCs before implantation [10] to maximize cell retention and viability and, in the case of bone engineering, to improve vascular network formation [1]. Furthermore, there are still some concerns about biosafety and efficacy of MSCs for clinical applications [11] as well as several other associated risk factors, such as the MSC differentiation status [12]. The unavailability of sufficient numbers of human primary cells is likely to delay the advance of research in these fields. This lack of primary MSCs is not only due to them being scarce (mainly healthy ones), but also the limited lifespan of the cells after isolation and in vitro culture. It has been described that human MSCs can achieve a maximum of 30–40 population doublings (PDs) in vitro before they lose their proliferation potential [13,14,15]. In addition, heterogeneity increases between MSCs which have been derived from the same donor at different passages, and the expanded MSCs progressively lose their differentiation potential [7,8,13]. There is also variability among donors [16], apart from the single MSC-derived clones isolated from the same donor [17,18,19]. For these reasons, human cell lines, and specifically iMSC lines, are only being used for research purposes.

Nowadays, a high number of MSC lines that display specific characteristics and differentiation capabilities have been generated and are valuable tools as part of models of disease and tissue repairing strategies. Different MSC lines have been employed for testing [17,20,21,22,23,24,25] or producing [26,27] engineered scaffolds for skeletal applications, and for both investigating the MSC differentiation process [28,29,30,31,32,33,34] and finding new ways to improve it [35,36,37,38,39]. Additionally, these cell lines have also been used for analyzing functional makers [19,40] or even for exploring their roles in different diseases, such as osteoarthritis [41,42].

The aim of this review was to analyze the characteristics of the MSC lines that are being used currently. Also, we aimed to investigate whether the MSC lines keep the phenotype of the primary cells from which they were derived, and if they could indeed be good models of tissue regeneration and disease.

## 2. Methodology

This review was carried out by employing Web of Science, Scopus, and PubMed databases from 1st January 2015 to 30th September 2019. In order to identify the human immortal MSC (iMSC) lines that are being used currently in the fields of bone and cartilage research, the keywords used were ALL = (mesenchymal AND (“cell line” OR immortal*) AND (cartilage OR chondrogenesis OR bone OR osteogenesis) AND human). Only original research studies were included in the analysis. References of the selected articles were included when relevant, and duplicates were excluded. After screening the title/abstract or full text, articles in which no human iMSC line was employed for osteogenesis or chondrogenesis experiments were excluded. The PRISMA flow diagram [43] is shown in Figure 1. This way, we identified 38 human iMSC lines derived from MSCs of single or “pooled” donors and whose osteogenic and/or chondrogenic potential had been tested.

For each iMSC line identified, we collected the following data: immortalization genes and method employed; tissue of origin and donor characteristics; if the iMSC line was clonal; whether it was tumorigenic and how its tumorigenicity had been assayed; and if it had been validated by short tandem repeat (STR) genotyping, as seen in Table 1. In addition, we investigated how its multidifferentiation (osteogenic, chondrogenic, and adipogenic) potential had been assessed and what results were obtained, as seen in Table 2. Afterwards, we described the immortalization strategies and their mechanism of action, as well as the outcomes and the characteristics of the iMSC lines revised, focusing on their osteogenic and chondrogenic capacities and potential usefulness for bone and cartilage regeneration research.

## 3. Immortal Mesenchymal Stromal Cell (iMSC) Lines

### 3.1. Immortalizing Human Adult MSCs

Immortalization is the process by which cells acquire an unlimited proliferation potential by bypassing senescence [106]. There are two types of senescence that cells must evade for their immortalization: replicative senescence, caused by telomere shortening and resulting chromosomal instability, and nonreplicative senescence, promoted by cellular stress, DNA damage, or oncogenic signals.

The initial in vitro growth arrest of human primary MSCs is presumed to be due to nonreplicative senescence, which is regulated by p53 and Rb-related pathways. In response to stress, the tumor suppressor p53 is phosphorylated and liberated from its binding to E3 ubiquitin ligase Mdm2, hence activating the senescence pathways. During quiescence, unphosphorylated Rb proteins control cell proliferation by binding and inhibiting E2F transcription factors, thus blocking cell cycle progression. During cell growth, signaling pathways that phosphorylate Rb proteins are activated, promoting its disassociation from E2F and allowing for the expression of E2F-dependent genes necessary for cell division [107,108]. Inhibition of p53 and inactivation of Rb by viral oncogenes have been shown to extend the life span of several cell types in culture, but telomeres maintenance is also needed for preventing replicative senescence [106]. Since human primary MSCs undergo progressive telomere shortening during serial passaging, human telomerase reverse transcriptase (hTERT) expression is needed to avoid telomere shortening [70]; otherwise, telomeres will shorten with every cell division until a critical threshold at which cells enter senescence.

Both simian virus 40 large T antigen (SV40LT) and human papillomavirus (HPV) E6/E7 gene transduction promote cell cycle progression by interfering with p53 and Rb-mediated pathways. SV40LT binds to these two proteins, thus releasing the activity of E2F transcription factors and avoiding growth arrest [107]. HPV E6/E7 proteins work in a similar manner, with p53 being the principal target of E6 and Rb being degraded via the ubiquitin proteasome pathway by the action of E7 [109]. SV40LT transduction has been employed for immortalizing MSCs derived from bone marrow from young and old donors [45,47], umbilical cord [49], cranial periosteum [51], coronal sutures [50], dental follicle [52], peripheral blood [48], and osteoarthritic cartilage [41]. SV40LT expression increases the lifespan of MSCs and usually raises its proliferation rate as well, but it has also been observed that this increased lifespan is not unlimited. Lee et al. (2015) reported that after more than 80 passages, SV40LT-transduced MSCs decreased their growth rate and entered senescence, indicating that this antigen is not enough for complete immortalization of MSCs [44].

HPV E6/E7 genes have also been used for immortalizing MSCs derived from bone marrow [54,58], and, in a similar way, E6/E7-transduced MSCs have been reported to enter a period of growth arrest after 70 PDs, suggesting a limited effect of E6/E7 in prolonging lifespan [58]. The same occurs with the p16 antagonist Bmi1, which has been reported to extend the lifespan of ligament-derived MSCs [62], but also to be insufficient for immortalization, with Bmi1-transduced adipose tissue-derived MSCs entering senescence after 55–60 PDs [110]. Since neither SV40LT nor E6/E7 proteins can promote the telomere maintenance needed to achieve an unlimited lifespan, SV40LT and E6/E7-transduced MSCs reported as immortal MSCs must have acquired a mechanism to prevent telomere shortening, or they will eventually undergo replicative senescence. Nevertheless, short telomeres are a source of chromosomal instability, and, if p53 activity is inhibited by SV40LT or E6/E7 proteins, alterations resulting from this instability will increase the mutability of the genome and might eventually give rise to telomerase re-expression [106].

It has been stated that hTERT transduction allows senescence evasion while maintaining in vitro and in vivo osteogenic ability of MSCs [63,91]. Transduction of hTERT alone has been employed to generate iMSC lines, but, since hTERT has no effect over non-replicative senescence, it has also been reported to fail to immortalize MSCs derived from bone marrow [13,18,58,74,104] and adipose tissue [85]. Skårn et al. (2014) described that only one out of nine hTERT-transduced bone marrow-derived MSC clones was able to proliferate over 40 PDs, and even this clone showed a slow proliferation rate similar to that of primary MSCs [71]. Other authors have confirmed that hTERT-transduced MSCs displayed a lifespan similar to that of primary MSCs (about 30–40 PDs) [18,81]. Okamoto et al. (2002) observed that p16 expression was upregulated in hTERT-transduced MSCs during passaging, finally leading to senescence despite maintenance of telomeres length. Conversely, MSCs transduced with both hTERT and E6/E7 were able to proliferate during more than 80 PDs [18], overcoming senescence, in the same way that MSCs transduced with both hTERT and SV40LT acquired an unlimited lifespan [13,85,110].

However, in addition to giving rise to MSC lines derived from bone marrow from young and/or healthy donors [15,63,71], which may well be less prone to suffer nonreplicative senescence, hTERT transduction has also been shown to immortalize placenta-derived MSCs of fetal and maternal origin [91], umbilical cord-derived MSCs [86], periodontal ligament-derived MSCs [89], adipose tissue-derived MSCs [84], and, importantly, osteoarthritic cartilage-derived MSCs [94]. Therefore, whether hTERT is sufficient to immortalize MSCs remains controversial. Immortalization requirements could be dependent on cell characteristics, cell culture conditions, and any factors that influence proneness to nonreplicative senescence, and seems not to be related to tissue of origin.

Transduction of hTERT has also been employed in combination with E6/E7 [18,55,58,85,101,104] or SV40LT [13,85,110]. This combination of genes leaded to an unlimited proliferation potential of the cells, which could not be obtained with the transduction of one gene only [58,85]. In the study of Balducci et al. (2014), the combination of hTERT and SV40LT was more efficient than hTERT and E6/E7 in improving growth rate of adipose-tissue-derived MSCs, but both combinations were efficient in overcoming senescence [85]. In short, hTERT alone, SV40LT alone, or E6/E7 alone could be enough to achieve immortalization of MSCs or not, but the combination of p53/Rb repression together with a mechanism of telomeres maintenance has always proven to be successful.

### 3.2. Multidifferentiation Potential of iMSCs

#### 3.2.1. Osteogenic Potential

Out of the 35 iMSC lines included in this review whose osteogenic potential had been tested, only one line showed no mineralization ability. This hASCs-T cell line was derived from adipose tissue and immortalized with a combination of hTERT and SV40LT [85]. All the resting iMSCs were capable of osteogenically differentiating upon induction, as shown by the standard histochemical staining—alizarin red staining (ARS), Von Kossa staining (VKS), and alkaline phosphatase staining (APS)—and gene expression analysis of bone-related genes RUNX2, osteocalcin, alkaline phosphatase (ALP), BMP-2, bone sialoprotein (BSP), and COL1A1, as seen in Table 2.

In four out of the thirty iMSC lines which displayed osteogenic potential, this differentiation ability was increased in comparison with primary MSCs, in terms of mineralization [14,85] and osteogenesic-related gene expression [44,51]. This increment of the osteogenic potential neither seems to be related to tissue’s origin, since it has been detected in iMSCs from bone marrow [14,44], cranial periosteum [51], and adipose tissue origin [85], nor seems to be related to immortalization protocol, as two of these cell lines were transduced with SV40LT [44,51], one was transduced with hTERT alone [14] and the other one with hTERT and E6/E7 [85].

On the other hand, there were four iMSC lines that showed reduced osteogenic potential in comparison with primary MSCs, all of them transduced with hTERT alone: BMA13H, hASCs-T, GB/hTERT MSCs, and Pelt cells. Two of these cell lines, BMA13H and hASCs-T, were incompletely immortalized [74,85], and so in these cases the reduction of osteogenic potential could be a result of the progressive loss of differentiation potential that occurs in primary MSCs as well. Also, nonreplicative senescence effects cannot be discarded in the reduction of osteogenic potential observed in GB/hTERT MSCs (derived from umbilical cord) and Pelt cells (derived from the periodontal ligament) [86,90].

Of note, all the reviewed MSC cell lines immortalized with hTERT and E6/E7 were described to maintain or enhance their osteogenic potential. Interestingly, the hTERT and E6/E7-transduced iMSC line 3A6 showed a higher osteogenic potential than their E6/E7-only-transduced counterpart, the KP cells [55], suggesting that a complete immortalization could be beneficial for the bone-forming capacity of MSCs. Nonetheless, hTERT expression in MSCs has also been shown to upregulate osteogenesis-related genes such as RUNX2, osterix, and osteocalcin [91], and SV40LT-transduced iMSCs have shown higher levels of RUNX2 than primary MSCs without any osteogenic stimuli as well [51]. It is commonly accepted that osteogenesis is the default differentiation pathway for MSCs [15,111] and the most commonly retained differentiation lineage at later passages. Thus, we hypothesize that this high expression of bone-related transcription factors might be due to an osteogenic commitment of later passaged MSCs instead of being due to an effect of the transduction of the immortalization genes. These signs of “spontaneous differentiation” have also been observed in the KP cells, but have been lost by complete immortalization of these cells with hTERT in addition to E6/E7 genes [55].

A reduction of osteogenic potential was also observed in some MSC cell lines immortalized with hTERT and SV40LT, such as hASCs-TS [85]. Song et al. (2017) also pointed out that reversibly immortalized iSuPs (derived from coronal sutures) presented increased osteogenic potential when the immortalization gene SV40LT was removed [50]. Shu et al. (2018) have proposed that MSCs with higher proliferative activity, such as SV40LT-transduced MSCs, may need a longer time to differentiate towards the osteogenic lineage [49]. Tátrai et al. (2012) reported that adipose tissue-derived MSCs immortalized with a combination of SV40LT and hTERT showed a higher growth rate, as a well as a reduced osteogenic and adipogenic potency [110]. The literature shows that the osteogenic differentiation potential is the most commonly retained path after immortalization, and that MSCs are able to differentiate towards this lineage if they are completely immortalized and adequate times and strategies of osteogenic induction are used.

#### 3.2.2. Chondrogenic Potential

Only 23 iMSC lines out of the 38 included in this review have been submitted for analysis of their chondrogenic potential. Two iMSC lines did not show any chondrogenic potential when assessed in two-dimensional culture: UE6E7T-2, derived from bone marrow and transduced with E6/E7 and hTERT [99]; and iSuPS, derived from coronal sutures and transduced with SV40LT [50]. Three other iMSC lines (SCP-1, BMA13H, and 3 Hits hMPC), all derived from bone marrow-MSCs, showed reduced chondrogenic potential in comparison to primary cells; all these iMSC lines were chondrogenically induced in three-dimensional culture and contained hTERT as immortalization gene, and two of them were incompletely immortalized [14,74,101].

In many cases, the chondrogenic potential of iMSCs has been scarcely analyzed by one single histochemical staining, either alcian blue staining (ABS) or toluidine blue staining (TBS) [14,54,55,71,91,101]. Although chondrogenic transcription factor Sox9 and type II collagen upregulation have been detected in chondrogenic-induced iMSCs, these cells showed the same proneness to hypertrophy as primary MSCs, with type X collagen expression [15,35] and low-quality cartilage production [22] with scarce exceptions [41,94], even when three-dimensional culture was performed [15,22,35]. This low-quality cartilage generation is relatively common among MSCs. It has been proposed that these cells are intrinsically committed to bone formation through the endochondral ossification pathway, and that they follow this differentiation program after being exposed to chondrogenic stimuli [111]. However, the lack of reproducibility among chondrogenic protocols is also a feasible explanation [112], and it is important to note that good results have been obtained when performing chondrogenesis onto suitable scaffolds [95]. Finger et al. (2003) found that immortalized cell lines obtained from chondrocytes were highly proliferative and showed less expression of genes involved in matrix synthesis and turnover than expected [113], in the same way that higher proliferation rates are related to reduced mineralization upon osteogenic induction [49,110].

Despite this, it has been shown that low chondrogenic iMSCs can stimulate chondrocyte differentiation when cocultured [71], possibly through the same trophic effects as primary MSCs. It is important to take into account that MSCs exists as heterogeneous populations, and that the iMSCs differentiation properties could be derived from primary cells or be related to the clonal selection, as has been proposed by Bourgine et al. (2014) [15]. Therefore, the differentiation potential of iMSCs should be assessed in comparison with their untransduced counterparts—the same clone or the primary cells derived from the same donor—in order to detect variations as a consequence of immortalization. Selection of MSC subsets or development of methods to stimulate MSCs to induce and/or to modulate specific attributes of the cells could give rise to more chondrogenic iMSC lines [9].

#### 3.2.3. Adipogenic Potential

The adipogenic potential of 30 out of 38 iMSC lines was assessed, mainly by oil red O staining (OROS), but also by gene expression analysis of adipogenesis-related genes, such as transcription factor PPARγ, with different results, as seen in Table 1. Several hTERT-transduced iMSC lines derived from the bone marrow [19,101], placenta [91], and adipose tissue [85] showed a reduction in adipogenic potential in comparison with primary MSCs. In addition, the bone marrow-derived KM101 iMSC line was unable to differentiate towards the adipogenic lineage [45], similar to incompletely immortalized hASCs-T [85]. Moreover, adipogenic potential of the 3A6 iMSC line was reduced in comparison with the KP cells from which they were derived, unlike osteogenic potential [55]. Conversely, all the reviewed MSC cell lines immortalized with SV40LT maintained their adipogenic potential. The adipose-tissue-derived ASC/TERT1 cell line showed an increase in adipogenic potential after immortalization with hTERT [22]. Once more, iSUPS differentiation potential towards the adipogenic lineage was increased when SV40LT was removed [50].

### 3.3. Surface Markers Expression of iMSCs

In 2006, the International Society for Cell Therapy proposed a panel of cell surface markers to identify human MSC, including CD73, CD90, and CD105 [114]. However, none of these markers are specific for MSCs, and their expression does not imply a multidifferentiation ability [115], since the same expression pattern can be found in other cell types, such as fibroblasts [116]. Although the expression of these surface markers is usually investigated in primary MSCs before immortalization, none of the articles performed sorting selection; instead, the whole isolated population or uncharacterized clones were employed for transduction. Nevertheless, the level of expression of these makers may change due to passaging and culture conditions [81,115], and their expression in primary cells does not guarantee that they will be expressed in immortalized cells, even if they are previously sorted.

For example, Abarrategi et al. (2018) noticed a lowering of CD73 and CD105 expression in iMSCs in comparison with primary MSCs [102], and Alexander et al. (2015) observed that cranial periosteum-derived TAg cells were less CD105-positive, but more CD146-positive than primary cells [51]. Adipose-tissue-derived hASCs-TS and hASCs-TE showed the same decrease in CD105 expression and an increase in CD146 expression [85]. On the contrary, hTERT-transduced BMA13H maintained CD105, but displayed reduced CD90 expression [81]. It is not possible to know if these changes were caused by transduction of immortalization genes, subculturing, or both.

In addition, it is not clear whether the expression of a certain surface marker of this traditional panel confers an advantage to differentiate towards a specific lineage; for example, it has been shown that there are no differences in chondrogenic potential of MSCs caused by CD105 expression [117]. Therefore, the value of these surface markers for iMSCs characterization or selection may be limited. However, the expression of several surface markers outside this panel may have functional characteristics. For example, James et al. (2015) noted that nondifferentiating iMSC clones were uniquely CD137-positive [19], and Jayasuriya et al. (2018) pointed that SV40LT-transduced OA-MSCs expressed high levels of CD54, which is lowly expressed by bone marrow-derived MSCs but constitutively expressed by articular chondrocytes [41].

### 3.4. Clonality, Selection and Validation

It is known that polyclonal expansion favors selection of faster growing cells, while clone characterization and selection may enable the maintenance of a subpopulation of cells with more desirable characteristics [81]. Variations in differentiation potential exists among clones derived from one single donor [19], and even among subclones derived from single MSCs [84]. Therefore, careful selection of clones could favor certain applications.

Half of the iMSC lines that are reviewed here were clonal, while the other half were nonclonal. However, some clones were not selected by their differentiation abilities or surface markers expression profile, but were randomly picked [91] or chosen because of their proliferation capacity [71]. On the contrary, Bourgine et al. (2014) decided to select the clone with the most prominent osteogenic differentiation capacity, thus obtaining an iMSC line likely to be suitable for bone regeneration approaches, but still with weak chondrogenic potential. Further studies are needed to know if the use of a similar approach would give rise to an iMSC line with better chondrogenic potential [15].

Of note, only eight of the iMSC lines reviewed here—four of them derived from the same donor—have been submitted to short tandem repeat (STR) analysis to confirm whether they originated from one particular donor. This validation is important, especially if these iMSC lines are employed for basic research about MSCs biology; for instance, comparing cell lines originating from young and elderly donors or investigating the characteristics or behavior of these cells in skeletal diseases. Using this approach, Jayasuriya et al. (2018) generated and analyzed clonal iMSC lines from knee articular cartilage of osteoarthritic patients, identifying the existence of two MSC populations in human osteoarthritic cartilage; one preferentially undergoing chondrogenesis and the other exhibiting higher osteogenic potential. In this case, the generated cell lines were properly submitted to STR genotyping [41].

### 3.5. Tumorigenicity

MSCs have been described as resistant to malignant transformation, requiring the combination of several events to achieve an oncogenic phenotype. Primary MSCs have been widely used in clinical trials, but immortalized MSCs transduced with proto-oncogenes can eventually become tumorigenic, making them useless for clinical approaches, but not for research purposes. The tumorigenicity of 18 out of 38 iMSC lines included in this review was investigated by either soft agar colony formation assay or in vivo tumorigenicity test in immunodeficient mice (IDM). There were only two cases in which these cells showed signs of tumorigenicity; c-Fos-transduced 3 Hits hMPCs [102] and high passaged UE6E7T-3 [96], as seen in Table 1. In the case of 3 Hits hMPCs, it is not surprising that the transduction of the proto-oncogene c-Fos led to oncogenic transformation of iMSCs already transduced with hTERT and E6/E7 genes. Importantly, transformed iMSCs lost their phenotype and experienced changes in their differentiation potential, with c-Fos-transformed iMSCs showing reduced adipogenic and osteogenic potential and a conserved ability to specifically differentiate towards the chondrogenic lineage, as well as forming chondrogenic tumors in IDM. However, 3 Hits hMPCs did not display tumorigenic features despite accumulating oncogenic mutations in hTERT and E6/E7 genes [102], thus confirming that iMSCs need further signals to initiate carcinogenesis [106].

However, oncogenic mutations may arise during passaging of iMSCs [118]. In this regard, culture conditions are important, since hTERT-transduced iMSCs seeded at low densities during long periods of time have been reported to be tumorigenic [70]. Low density seeding provides an advantage for clones with oncogenic mutations that display higher growth rates; the lower the density seeding, the faster the accumulation of these oncogenic clones in the population. hTERT and E6/E7-transduced UE6E7T-3 at high passages (252 PDs) were capable of forming colonies in soft agar and sarcomas in IDM, while lower passaged cells (less than 200 PDs) did not shown any sign of malignant transformation [96]. This may indicate that although iMSCs have an unlimited lifespan, their ability to maintain their phenotype may be restricted to the first PDs, and that their characteristics should be submitted to periodical testing.

It is important to note that hTERT expression also has a role in the achievement of cellular capacities related to tumorigenesis, such as angiogenesis and immune system evasion. In the same way, the inactivation of p53 and Rb by SV40LT and E6/E7 proteins is related to the acquisition of cancer-related features. Moreover, if hTERT is re-expressed after SV40LT or E6/E7 transduction, its recovery could favor the fixation of aberrant karyotypes that lead to malignant phenotypes [106]. Unsurprisingly, the introduction of immortalization genes in MSCs alters the expression levels of genes associated with stem cell functions [91], which highlights the need for detailed characterization of iMSC lines.

### 3.6. In Vivo Bone Formation Capacity

One of the fundamental characteristics of MSCs is their ability to form ectopic “ossicles” which mimic the architecture of bone marrow [119]. The ectopic bone formation ability of hTERT-transduced MSCs has been investigated in IDM. Simonsen et al. (2002) found that hMSCs-hTERT generated bone-enclosing bone marrow cells and adipocytes when implanted subcutaneously with hydroxyapatite/tricalcum phosphate powder [63]. hMSCs-hTERT-derived clones formed ectopic bone marrow stroma-supporting hematopoiesis and adipocytes after in vivo transplantation as well [70]. However, after extensive subculturing, one of these clones was found to produce tumors, composed mostly of mesoderm-type cells [70]. Bourgine et al. (2014) also assessed the bone formation ability of MSOD together with ceramic granules in a fibrinogen/thrombin gel. They found that MSOD secreted a dense collagen matrix and formed osteoid tissue [15].

## 4. Conclusions

A number of iMSC lines have been generated in an attempt to overcome the limitations associated with primary MSCs. These cell lines have had many in vitro applications, including testing of engineered scaffolds for bone and cartilage repair, decellularized extracellular matrix production, investigation of the MSCs differentiation process at the molecular level, optimization of the current differentiation protocols, and analysis of their behavior in the pathological joints. However, the application of these cells has been for research purposes only as they present a risk of tumorigenicity.

Several approaches have been employed to confer an unlimited proliferation potential to MSCs, mainly involving viral genes and hTERT transduction, with different degrees of success. It is still unclear which set of genetic alterations are necessary and sufficient for MSC immortalization, but it probably involves abrogation of replicative and nonreplicative senescence.

Alterations of the multidifferentiation potential of MSCs after immortalization have been described, with the osteogenic potential being the best conserved in fully immortalized MSC lines. However, there are also iMSC lines capable of differentiating towards the chondrogenic lineage when cultured in 3D environment. In addition, some studies suggest that other characteristics of iMSC lines, such as secretion of trophic factors, are maintained in MSCs despite immortalization.

The ideal iMSC line should retain the phenotypic and functional characteristics of MSCs, as well as a normal karyotype. The usefulness of these cell lines in bone and cartilage regenerative medicine research would be increased if clones were carefully selected and validated. Before employing iMSCs for basic or applied bone and cartilage research, their characteristics should be fully understood. Moreover, the maintenance of these characteristics should be assessed periodically throughout passaging, as immortalization does not guarantee it, and both polyclonal expansion and low-density seeding should be avoided to prevent malignant transformation. If these requirements are fulfilled, iMSC lines could be useful and convenient tools for basic research, testing of tissue engineering approaches, and production of biotechnological products, among other applications.

## Figures and Tables

**Figure 1 ijms-20-06286-f001:**
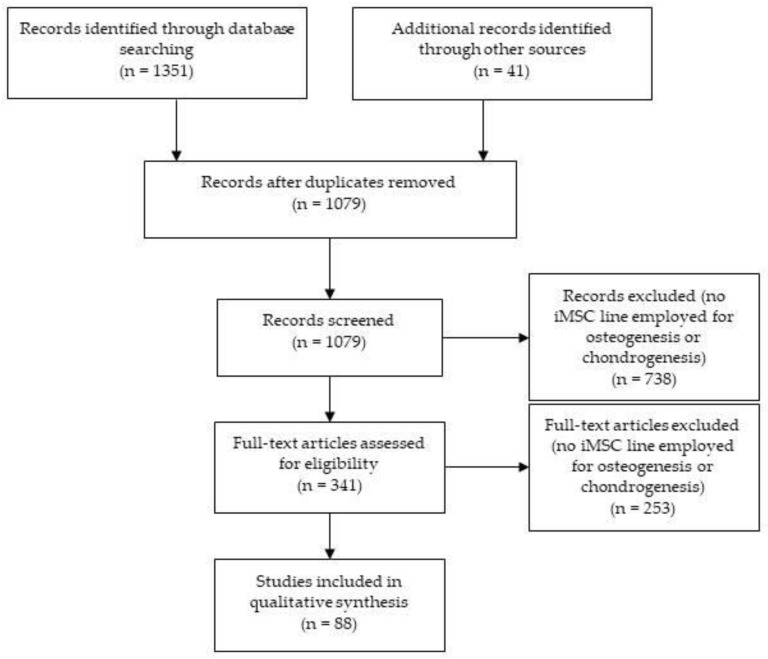
PRISMA flow diagram showing the search process carried out to select articles to be analyzed for this review.

**Table 1 ijms-20-06286-t001:** Basic characteristics of the reviewed immortal MSC (iMSC) lines.

MSC Line	Immortalization Genes	Immortalization Method	Tissue	Donor Characteristics	STR Genotyping	Clonality	Tumorigenicity	References
hMSC-T	SV40LT	Transfection	Bone marrow	Unknown	No	Unclear ^1^	No (tested by soft agar)	[44]
KM101	SV40LT	Transfection	Bone marrow	48-year-old male	No	Yes	Not tested	[45,46]
L87/4	SV40LT	Transfection	Bone marrow	70-year-old male	No	Yes	Not tested/shown	[17,47]
V54/2	SV40LT	Transfection	Peripheral blood	Healthy donor	No	Yes	Not tested/shown	[17,48]
iUC-MSCs	SV40LT	Retroviral transduction	Umbilical cord	Unknown	No	No	No (tested in IDM)	[49]
iSuPs	SV40LT	Retroviral transduction	Coronal sutures	15 to 17-month-old males	No	No	No (tested in IDM)	[50]
TAg cells	SV40LT	Lentiviral transduction	Cranial periosteum	Healthy (fracture patient)	No	No	Not tested	[51]
iDFCs	SV40LT	Retroviral transduction	Dental follicle	Three young adults (18–20 years old)	No	Yes	Not tested	[52,53]
OA-MSCs	SV40LT	Retroviral transduction	Articular cartilage (knee)	Osteoarthritic 61-year-old male and 69-year-old female	Yes	Yes ^2^	Not tested/shown	[41,42]
KP	E6/E7	Retroviral transduction	Bone marrow	61-year-old female	No	No	No (tested in IDM)	[54,55,56,57]
UE6E7-16 ^3^	E6/E7	Retroviral transduction	Bone marrow	91-year-old female	Yes	Yes	Not shown	[58,59]
HS-27	E6/E7	Retroviral transduction	Bone marrow	Adult donor	No	Yes	Not tested	[60,61]
PDLSC-Bmi1 ^4^	Bmi1	Retroviral transduction	Periodontal ligament	15 to 20-year-old donors	No	No	Not tested	[62]
hMSC-hTERT	hTERT	Retroviral transduction	Bone marrow	Healthy 33-year-old male	No	No ^5^	No (tested in IDM)	[30,33,34,63,64,65,66,67,68,69]
TERT4 (hMSC-hTERT derived)	hTERT	Retroviral transduction	Bone marrow	Healthy 33-year-old male	No	No	No (tested in IDM) ^6^	[35,40,70]
iMSC#3	hTERT	Retroviral transduction	Bone marrow	Healthy male	No	Yes	No (tested in IDM)	[71,72,73]
BMA13H ^7^	hTERT	Retroviral transduction	Bone marrow	Unknown	No	No	Not tested	[74,75]
SCP-1	hTERT	Lentiviral transduction	Bone marrow	Unknown	No	Yes	No (tested in IMD and by soft agar assay)	[14,20,24,25,38,76,77,78,79,80]
Y201	hTERT	Lentiviral transduction	Bone marrow	Unknown	No	Yes	No (tested in IDM)	[19,36,81,82,83]
Y101	hTERT	Lentiviral transduction	Bone marrow	Unknown				[19,32]
MSOD	hTERT	Lentiviral transduction	Bone marrow	Healthy 55-year-old female	Yes	Yes	No (tested in IDM)	[15,26]
ASC/TERT1	hTERT	Retroviral transduction	Adipose tissue	Unknown	Yes	No	No (soft agar assay)	[22,84]
hASCs-T ^7^	hTERT	Lentiviral transduction	Adipose tissue	Two males and two females (21 to 59 years old)	No	No	No (soft agar assay)	[85]
GB/hTERT MSCs	hTERT	Transfection	Umbilical cord	Unknown	No	No	No (soft agar assay)	[86]
SDP11	hTERT	Transfection	Dental pulp	6 to 8-year-old donors	No	Yes	Not tested	[87,88]
Pelt cells	hTERT	Retroviral transduction	Periodontal ligament	Adult donor	No	No	Not tested/shown	[31,89,90]
CMSC29	hTERT	Retroviral transduction	Placenta (Chorionic Villi)	Unknown	No	Yes	No (tested by soft agar assay)	[91,92,93]
DMSC23	hTERT	Retroviral transduction	Placenta (Decidua Basalis)	Unknown	No	Yes	No (tested by soft agar assay)	[91,92,93]
CPC531	hTERT	Lentiviral transduction	Articular cartilage (knee)	65 to 75-year-old patients	No	Unclear ^1^	Not tested/shown	[94,95]
hASCs-TS (same parental cells as hASCs-T)	hTERT and SV40LT	Lentiviral transduction	Adipose tissue	Two males and two females (21 to 59 years old)	No	No	No (soft agar assay)	[85]
3A6 (KP-derived)	hTERT and E6/E7	Transfection (hTERT)	Bone marrow	61-year-old female	No	Yes	Not tested	[39,55,56]
hASCs-TE (same parental cells as hASCs-T)	hTERT and E6/E7	Lentiviral transduction	Adipose tissue	Two males and two females (21 to 59 years old)	No	No	No (soft agar assay)	[85]
UE6E7T-3 (same parental cells as UE6E7-16)	hTERT and E6/E7	Retroviral transduction	Bone marrow	91-year-old female	Yes	Yes	Tested in soft agar at “low” (PDs ≤ 200) and high (PDs = 252) passages, with only high passage UE6E7T-3 being capable of forming colonies; high passage UE6E7T-3 formed sarcomas in IDM	[28,96,97]
UE6E7T-11 (same parental cells as UE6E7-16)	hTERT and E6/E7	Retroviral transduction	Bone marrow	91-year-old female	Yes	Yes	Not shown	[58,98]
UE6E7T-2 (same parental cells as UE6E7-16)	hTERT and E6/E7	Retroviral transduction	Bone marrow	91-year-old female	Yes	Yes	Not shown	[99]
imhMSCs	hTERT and E6/E7	Retroviral transduction	Bone marrow	Unknown	No	Unclear ^1^	No (tested in IDM)	[18,23,29,100]
3 Hits hMPC	hTERT and E6/E7	Retroviral transduction	Bone marrow	Healthy 34-year-old male	Yes	No	No (tested in IDM; only c-Fos-transduced cells were tumorigenic)	[101,102,103]
UE7T-13 (same parental cells as UE6E7-16)	hTERT and E7	Retroviral transduction	Bone marrow	91-year-old female	Yes	Yes	Not shown	[21,37,88,104,105]

^1^ Clones have been generated but it is unclear if they have been used afterwards. ^2^ Several clones have been generated and analyzed. ^3^ Not completely immortalized because E6/E7 was sufficient to extend lifespan but not to bypass senescence. ^4^ Probably incompletely immortalized. ^5^ [30,68,69] employed hMSC-hTERT-derived clones. ^6^ TERT20 at higher passages formed tumors composed of mesoderm type cells in IDM. ^7^ Not completely immortalized because hTERT was not sufficient to bypass senescence.

**Table 2 ijms-20-06286-t002:** Differentiation potential of reviewed iMSC lines.

MSC Line	Osteogenic Potential	Chondrogenic Potential	Adipogenic Potential
hMSC-T	Positive for VKS and osteocalcin upregulation (increased compared with primary MSCs) [44]	Not tested	Not tested
KM101	Positive for ALP activity [46]	Not tested	Tested and no adipogenic differentiation potential was found (also not shown) [45]
L87/4	Not tested/shown	Positive for ABS and ColII immunostaining in 3D alginate and PGA/PLLA scaffolds [17]	Not tested/shown
V54/2	Not tested/shown	Positive for ABS and ColII immunostaining in 3D alginate and PGA/PLLA scaffolds [17]	Not tested/shown
iUC-MSCs	Positive for Runx2 and Osteocalcin upregulation [49]	Positive for Sox9 upregulation [49]	Positive for PPARγ upregulation [49]
iSuPs	Positive for ARS (increased if SV40LT is removed) and osteogenesis-related genes upregulation [50]	No chondrogenic differentiation potential was found (also not shown) [50]	Positive for OROS (increased if SV40LT is removed) [50]
TAg cells	Positive for hydroxyapatite formation (showing earlier and stronger mineralization than parental cells) and upregulation of osteogenesis-related genes (increased compared with primary cells) [51]	Not tested	Not tested
iDFCs	Positive for ARS, APS, and osteogenesis-related genes upregulation; osteogenic potential similar to primary cells [52]	Positive for ABS and SOX9 upregulation in 2D culture [52]	Positive for OROS and adipogenesis-related genes upregulation (PPARγ and LPL) [52]
OA-MSCs	Positive for ARS and ALP upregulation [41,42]	Positive for SOS [41], ABS [42] and upregulation of Sox9, Col2A1, ACAN and COL10A1 [41,42] in pellet [41] and 2D culture [42]	Positive for OROS (weak staining) and LPL upregulation [41]
KP	Positive for APS, ARS, and VKS [54]	Proved by ABS [54,57] and ColII immunostaining [57] in pellet culture	Positive for OROS [54]
UE6E7-16	Positive for osteocalcin production [59]	Not tested/shown	Positive for PPARγ production [59]
HS-27	Positive for ALP activity, calcium deposition and osterix upregulation [61]	Not tested/shown	Positive for OROS in presence of steroids [60]
PDLSC-Bmi1	Positive for ARS, ALP activity, and osteogenesis-related genes upregulation [62]	Not tested	Positive for OROS [62]
hMSC-hTERT	Positive for ARS [64], ALP activity [33], upregulation of osteogenesis-related genes [33,63,68], and in vivo bone formation [63,68]	Positive for ABS [64] and ColII immunostaining [63,67] in 2D culture	Positive for OROS and upregulation of adipogenesis-related genes [33]
TERT4 (hMSC-hTERT derived)	Positive for ARS [40,70], ALP activity [40] and upregulation of osteogenesis-related genes [40]	Positive for ABS [35,70], GAG assay [35], and upregulation of ColII [35,70] but also ColX [35], in pellet culture; reduced compared with primary MSCs	Positive for OROS [40,70] and upregulation of adipogenesis-related genes [40]
iMSC#3	Positive for ARS, APS, and Runx2 upregulation [71]	Positive for ABS, TBS, and GAG assay in pellet culture [72]; low chondrogenic potential but stimulation of chondrocyte differentiation	Positive for OROS, adipogenesis-related genes upregulation [71,73], and NRS [73]
BMA13H	Positive for ARS (reduced compared with primary cells) [74]	Positive for ABS and GAG assay in 2D culture [74]; also positive for TBS, PSR and aggrecan and ColII immunostaining in 3D culture [75]; chondrogenic potential reduced compared with primary cells [74]	Positive for OROS (reduced compared with primary cells) [74]
SCP-1	Positive for VKS (increased compared with MSCs) [14], ARS [78], ALP activity [78], and upregulation of osteogenesis-related genes [76]	Positive for TBS in pellet culture [14]; ColII and GAG production in 3D printed scaffolds [24]	Positive for OROS [14]
Y201	Positive for ARS [19], ALP activity [19,82,83], and Runx2 upregulation [19,82]	Positive for ABS, GAG assay, and Sox9 upregulation in pellet culture [19]	Positive for OROS (reduced compared to primary MSCs) and upregulation of adipogenesis-related genes [19,82]
Y101 (derived from the same donor than Y201)	Proved by ARS [19,32], VKS [32], ALP activity, and osteogenesis-related genes’ upregulation [19,32]; osteogenic potential similar to Y201 [19]	Positive for ABS, GAG assay and Sox9 upregulation in pellet culture; chondrogenic potential similar to Y201 [19]	Positive for OROS and upregulation of adipogenesis-related genes; adipogenic potential reduced compared to Y201 [19]
MSOD	Positive for ARS [15], upregulation of osteogenesis-related genes [15,26], and in vivo bone formation [15]	Weak positivity for ABS and upregulation of ColX but not ColII nor Sox9, similarly to primary parental cells; tested in pellet culture [15]	Positive for OROS and PPARγ upregulation [15]
ASC/TERT1	Positive for VKS and ALP activity [84]	Positive for ABS, trichrome staining and ColII immunostaining in 3D scaffolds; reduced cartilage quality in comparison with chondrocytes [22]	Positive for OROS and PPARγ upregulation; adipogenic potential increased compared with primary cells [84]
hASCs-T	Positive for APS; reduced osteogenic potential in comparison with primary cells [85]	Not tested	Tested by OROS, but almost no lipid droplets detected [85]
GB/hTERT MSCs	Positive for ARS; reduced compared with primary cells [86]	Not tested	Positive for OROS [86]
SDP11	Positive for BMP-2 and ALP upregulation [88]	Not tested	Positive for OROS but not shown [88]
Pelt cells	Positive for ARS (slightly reduced compared with primary cells) [90] and cementogenesis-related gene expression [31]	Not tested	Not tested
CMSC29	Positive for ARS [91]	Positive for ABS in pellet culture [91,92]	Very weak positivity for OROS [91]
DMSC23	Positive for ARS (increased compared with CMSC29) [91,93]	Positive for ABS in pellet culture [91,92]	Very weak positivity for OROS [91]
CPC531	Positive for APS and upregulation of osteogenesis-related genes [95]	Spontaneous chondrogenesis in 3D alginate culture, proved by upregulation of ColII and Sox9 and downregulation of Runx2 and ColI [95]	Positive for OROS and upregulation of adipogenesis-related genes [95]
hASCs-TS (same parental cells as hASCs-T)	Tested by APS, but no mineralization detected [85]	Not tested	Positive for OROS; reduced adipogenic potential in comparison with primary cells [85]
3A6 (KP-derived)	Positive for ARS and VKS (increased compared with KP) [55], and also ALP activity [39]	Positive for ABS [55] and ColII upregulation [56] in pellet culture	Positive for OROS (reduced compared with KP) [55]
hASCs-TE (same parental cells as hASCs-T)	Positive for APS; increased in comparison with primary cells [85]	Not tested	Positive for OROS; slightly reduced in comparison with primary cells [85]
UE6E7T-3 (same parental cells as UE6E7-16)	Positive for ALP activity [97], ARS and upregulation of osteogenesis-related genes [28]	Not tested/shown	Positive for OROS [97]
UE6E7T-11 (same parental cells as UE6E7-16)	Positive for APS and bone sialoprotein (BSP) upregulation [98]	Not tested/shown	Not tested/shown
UE6E7T-2 (same parental cells as UE6E7-16)	Not tested/shown	Tested by ABS in 2D culture; negative under employed conditions [99]	Not tested/shown
imhMSCs	Positive for VKS and upregulation of osteogenesis-related genes [18]	Weak positivity for ABS and with weak upregulation of chondrogenesis-related genes (similarly to primary parental cells); tested in pellet culture [18]	Positive for OROS and PPARγ upregulation [18]
3 Hits hMPC	Positive for ARS [102,103], APS [101,102] and Runx2 upregulation [102]	Positive for ABS [102,103] and TBS [101] in pellet culture, but reduced compared with primary MSCs	Positive for OROS [101,102,103], but reduced compared with primary MSCs
UE7T-13 (same parental cells as UE6E7-16)	Positive for ARS [37,88,105] and ALP activity [88]	Not tested/shown	Positive for OROS [37,88]

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
