# Peer review of "Usefulness of Mesenchymal Cell Lines for Bone and Cartilage Regeneration Research"

_ijms, 2019, doi:10.3390/ijms20246286_

Round 1

Reviewer 1 Report

The manuscript by M. Piñeiro-Ramil  et al, entitled “Usefulness of mesenchymal cell lines for bone and 2 cartilage regeneration research”. The work is timely and important, but some aspects need to be addressed to allow the readers to interpret the data and studies discussed.

Main concerns are described bellow.

1 – The main concern is of methodology, the authors follow some of the principles of a systematic review, but not all. To have a systematic review the authors need not only to follow the PRISMA methodology, but also to describe better their process in the abstract and in a methodology section and to extract and analyze the relevant data. Also, the authors should check other databases, like PubMed and Embase, to make sure that all the literature on iMSC is included.

1.1 - This partial adoption of the systematic review workflow reflects in the abstract, which needs revising as some sentences are not clear, and are written as if the authors confirmed the data from other studies and did not just revise them. For example “The purpose of this review was to analyze the characteristics of the available iMSC lines and to examine the maintenance of the phenotype…” when the authors do not analyze the cell lines themselves, and instead revise the literature available on the characteristics of iMSC lines. Also, the authors should mention in the abstract the methods used for their systematic review.

The authors should complete and revise the manuscript either as a systematic review or as a descriptive review.

2 – There are a few aspects that the authors revise, but then do not conclude if there is, or there is not, sufficient evidence to support it, please clarify the following:

2.1 – It is unclear if there is evidence of the impact of immortalization on osteogenic differentiation.

2.2 – It is unclear if chondrogenic differentiation was tested in pellet cultures or in simple 2D cultures, which can condition the results obtained.

2.3 – It is unclear if MSC were selected based on stem markers

3 – Section “2.3. Surface markers expression of iMSCs”,  is not really focused on surface markers, their characterization and how they may, or not, help in selecting/characterizing iMSC, instead the authors seem to focus on iMSC selection strategies, talking about clonal selection and selection based in different iMSC characteristics. Please reformulate.

4 – In table 1, can the authors please include a column with the references for each line? Also, if the table will span over multiple pages the heading should be included in each page, to facilitate reading.

5 – On page 12, “Table 1. (continued)”, should be table 2 as the headings are not the same. So, table 1 would cover basic cell line characteristics and table 2, would cover their differentiation capacity. Also, instead of “proved by” perhaps the authors should say “positive for”

6 – Please revise the language mistakes, like the expression “human primary human cells”, which appears several times along the manuscript.

Author Response

REVIEWER 1

The manuscript by M. Piñeiro-Ramil et al, entitled “Usefulness of mesenchymal cell lines for bone and cartilage regeneration research”. The work is timely and important, but some aspects need to be addressed to allow the readers to interpret the data and studies discussed.

We would like to thank you for your comments.

Main concerns are described bellow.

1 – The main concern is of methodology, the authors follow some of the principles of a systematic review, but not all. To have a systematic review the authors need not only to follow the PRISMA methodology, but also to describe better their process in the abstract and in a methodology section and to extract and analyze the relevant data. Also, the authors should check other databases, like PubMed and Embase, to make sure that all the literature on iMSC is included.

The authors should complete and revise the manuscript either as a systematic review or as a descriptive review.

We performed the same search on Scoups and Pubmed to make sure that all the literature about iMSCs was included. Figure 1 and Tables 1 and 2 have been updated accordingly, and the new data found has been included in the manuscript. We have included a methodology section and a description of this methodology in the abstract as well.

1.1 - This partial adoption of the systematic review workflow reflects in the abstract, which needs revising as some sentences are not clear, and are written as if the authors confirmed the data from other studies and did not just revise them. For example “The purpose of this review was to analyze the characteristics of the available iMSC lines and to examine the maintenance of the phenotype…” when the authors do not analyze the cell lines themselves, and instead revise the literature available on the characteristics of iMSC lines. Also, the authors should mention in the abstract the methods used for their systematic review.

We have changed the abstract as suggested and included a description of the methods used for the review.

2 – There are a few aspects that the authors revise, but then do not conclude if there is, or there is not, sufficient evidence to support it, please clarify the following:

2.1 – It is unclear if there is evidence of the impact of immortalization on osteogenic differentiation.

Immortalization alters cell growth and thus the balance between cell growth and differentiation that must exist in a stem cell, but there is no sufficient evidence of it enhancing or reducing osteogenic potential of MSCs. We have clarified this point at the end of section “3.2.1. Osteogenic potential”.

2.2 – It is unclear if chondrogenic differentiation was tested in pellet cultures or in simple 2D cultures, which can condition the results obtained.

Table 2 has been updated including the culture method (2D or 3D) for chondrogenic differentiation. We also mention which culture methods were employed for each cell line in section “3.2.2. Chondrogenic potential”.

2.3 – It is unclear if MSC were selected based on stem markers

The expression of certain surface markers (like CD73, CD90 and CD105) were investigated in primary MSCs before immortalization, but the cells have not been selected by the expression of these markers before transduction, as it has been now stated in section “3.3. Surface markers expression of iMSCs”.

3 – Section “2.3. Surface markers expression of iMSCs”,  is not really focused on surface markers, their characterization and how they may, or not, help in selecting/characterizing iMSC, instead the authors seem to focus on iMSC selection strategies, talking about clonal selection and selection based in different iMSC characteristics. Please reformulate.

Text in section “3.3. Surface markers expression of iMSCs” has been revised to address these issues.

4 – In table 1, can the authors please include a column with the references for each line? Also, if the table will span over multiple pages the heading should be included in each page, to facilitate reading.

Table 1 has been changed as suggested.

5 – On page 12, “Table 1. (continued)”, should be table 2 as the headings are not the same. So, table 1 would cover basic cell line characteristics and table 2, would cover their differentiation capacity. Also, instead of “proved by” perhaps the authors should say “positive for”.

The second part of Table 1 has been converted in Table 2 as suggested. Also “proved by” has been changed to “positive for”.

6 – Please revise the language mistakes, like the expression “human primary human cells”, which appears several times along the manuscript.

These errors have been corrected.

Reviewer 2 Report

This manuscript reviews the potential application of immortalized human mesenchymal stromal cells (iMSCs) in bone and cartilage regeneration. This manuscript highlights the methods of immortalization and their effects on phenotype, tri-lineage differentiation potential (osteogenic, chondrogenic and adipogenic) and tumorigenicity of MSCs. Most of the iMSC lines maintained their phenotype and tri-lineage differentiation potential, and showed a low risk of tumorigenicity. Overall, the manuscript is interesting and well-written. However, there are a few things that the authors need to address.

1. hTERT was always used in conjunction with E6/E7 to completely immortalize MSCs, but not with SV40LT. Please discuss.

2. Please discuss how polyclonal expansion and low density seeding of iMSCs may lead to malignant transformation.

3. Have iMSCs been used for bone or cartilage regeneration in preclinical animal studies?

4. The authors should discuss the challenges of using iMSCs clinically in bone and cartilage regeneration.

5. The following sentences should be added into the section 2.2.1 Osteogenic potential.

All the reviewed MSC cell lines immortalized with hTERT and E6/E7 were found to maintain or enhance their osteogenic potential. A reduction of osteogenic potential was observed in some MSC cell lines immortalized with hTERT alone (g., BMA13H, hASCs-T, GB/hTERT MSCs, and Pelt cells) or hTERT and SV40LT (e.g., hASCs-TS).

6. The following sentence should be added into the section 2.2.3 Adipogenic potential.

All the reviewed MSC cell lines immortalized with SV40LT were found to maintain adipogenic potential.

7. Line 189: add “(Scp-1, BMA13H and 3 Hits hMPC)” after “iMSC lines”.

8. Line 191: replace “one” with “two”.

9. Line 218: replace “iMSC” with “hTERT-transduced iMSC”.

10. The following relevant works should be cited and discussed.

Biosafety and bioefficacy assessment of human mesenchymal stem cells: what do we know so far? (2018), Regenerative Medicine, 13 (2): 219-232. Risk factors in the development of stem cell therapy. (2011), Journal of Translational Medicine, 9 (1): 29.

Author Response

REVIEWER 2

Comments and Suggestions for Authors

This manuscript reviews the potential application of immortalized human mesenchymal stromal cells (iMSCs) in bone and cartilage regeneration. This manuscript highlights the methods of immortalization and their effects on phenotype, tri-lineage differentiation potential (osteogenic, chondrogenic and adipogenic) and tumorigenicity of MSCs. Most of the iMSC lines maintained their phenotype and tri-lineage differentiation potential, and showed a low risk of tumorigenicity. Overall, the manuscript is interesting and well-written. However, there are a few things that the authors need to address.

We would like to thank you for your comments. We think we need to emphasize in this manuscript that the use of iMSCs is only for research purposes and now we have included several sentences in the manuscript to make it clearer.  

hTERT was always used in conjunction with E6/E7 to completely immortalize MSCs, but not with SV40LT. Please discuss.

hTERT has been used much more often in combination with E6/E7 but it has also been used with SV40LT, as it is stated in the last paragraph of section 3.1. E6/E7 and SV40LT perform a similar function and they are “interchangeable”. None of the articles revised included a reason for choosing one or the other and we ignore why E6/E7 has been employed more often.

Please discuss how polyclonal expansion and low density seeding of iMSCs may lead to malignant transformation.

Polyclonal expansion favors selection of faster growing cells, as it has been stated in section "3.4. Clonality, selection and validation". Also, as it has been discussed in section “3.5. Tumorigenicity”, low density seeding provides an advantage for clones with oncogenic mutations, which display higher growth rates; the lower the density seeding, the faster the accumulation of these oncogenic clones in the population.

Have iMSCs been used for bone or cartilage regeneration in preclinical animal studies? and The authors should discuss the challenges of using iMSCs clinically in bone and cartilage regeneration.

Response points 3 and 4: iMSCs are not used for clinical purposes due to their tumorogeneicity risk. However, MSCs were widely used in clinical trials and in vivo animal studies, with very low success rates and other problematic issues (as shown in the literature recommended in the point 10). There is a necessity to obtain a deeper knowledge on the field to surpass this problematic, however the low number/accessibility of primary MSCs leads us to work with iMSCs. Animal studies were carried out to assess iMSCs ability to form ectopic bone in vivo, which is now discussed in section “3.6. In vivo bone formation capacity”. We have also included several sentences along the manuscript to make a clearer statement about the use of iMSCs only for research purposes.

The following sentences should be added into the section 2.2.1 Osteogenic potential.

All the reviewed MSC cell lines immortalized with hTERT and E6/E7 were found to maintain or enhance their osteogenic potential. A reduction of osteogenic potential was observed in some MSC cell lines immortalized with hTERT alone (g., BMA13H, hASCs-T, GB/hTERT MSCs, and Pelt cells) or hTERT and SV40LT (e.g., hASCs-TS). We have included these sentences throughout the section “3.2.1. Osteogenic potential”.

The following sentence should be added into the section 2.2.3 Adipogenic potential. All the reviewed MSC cell lines immortalized with SV40LT were found to maintain adipogenic potential.

This sentence has been included in section “3.2.3. Adipogenic potential”.

Line 189: add “(Scp-1, BMA13H and 3 Hits hMPC)” after “iMSC lines”.

It has been changed as suggested.

Line 191: replace “one” with “two”.

We have maintained “one” because we mean one single histochemical staining, ABS or TBS.

Line 218: replace “iMSC” with “hTERT-transduced iMSC”.

It has been changed as suggested.

The following relevant works should be cited and discussed.

Biosafety and bioefficacy assessment of human mesenchymal stem cells: what do we know so far? (2018), Regenerative Medicine, 13 (2): 219-232. Risk factors in the development of stem cell therapy. (2011), Journal of Translational Medicine, 9 (1): 29.

 The suggested works are very relevant and have been now cited in the introduction.

Round 2

Reviewer 1 Report

The authors addressed most of the concerns raised, but it would have been easier to analyse the revision if changes to the text were highlighted. Moreover, a few aspects should be improved further before publication. 

1)Table 1 is better, but its very long and not easy to understand. Can the authors please group the cell lines by common criteria, like for example the immortalization method and/or tissue of origin. That way it would be easier for the reader to understand the characteristics of the different cell lines. Also, can the authors please indicate why there are blank spaces, if it was not analyzed maybe it would be better to write not tested, as in table 2.

2) On p7, L264, the authors say that "...human MSC may be identified by expression of a specific panel of cell surface markers...", but on L267 state "However, taking into account that none of these markers are specific for MSCs..." Can the authors please clarify.

3) The grammar needs revising. Please see bellow some examples, but the text needs to be revised throughout.

Abstract: "After describing the immortalization protocol success, we had focus on the maintenance of the 28 parental phenotype and multipotency in iMSCs."should be "... we focused..." or "this review focused..."

p2, L79: "...articles in which none human iMSC line was employed..." should be "...no human iMSC line...".

p4, L151: "However, in addition to given rise to MSC lines..." should be "...giving rise..."

p9, L376: "...The usefulness of this cell lines in bone and cartilage..." should be these cell lines.

Author Response

The authors addressed most of the concerns raised, but it would have been easier to analyse the revision if changes to the text were highlighted. Moreover, a few aspects should be improved further before publication. 

1) Table 1 is better, but its very long and not easy to understand. Can the authors please group the cell lines by common criteria, like for example the immortalization method and/or tissue of origin. That way it would be easier for the reader to understand the characteristics of the different cell lines. Also, can the authors please indicate why there are blank spaces, if it was not analyzed maybe it would be better to write not tested, as in table 2.

We have group the cell lines for immortalization genes, and, after that, for tissue of origin and immortalization method. The blank spaces were due to different cell lines being derived from the same tissue and the donor, but this data have been included for each cell lines for clarification. The fact that different cell lines were derived from the same donor is indicated after cell line name. Table 2 has also been updated to the follow the same order established now in Table 1.

2) On p7, L264, the authors say that "...human MSC may be identified by expression of a specific panel of cell surface markers...", but on L267 state "However, taking into account that none of these markers are specific for MSCs..." Can the authors please clarify.

We have changed the beginning of section “3.3. Surface markers expression of iMSCs” to clarify this point.

3) The grammar needs revising. Please see bellow some examples, but the text needs to be revised throughout.

Abstract: "After describing the immortalization protocol success, we had focus on the maintenance of the 28 parental phenotype and multipotency in iMSCs."should be "... we focused..." or "this review focused..."

p2, L79: "...articles in which none human iMSC line was employed..." should be "...no human iMSC line...".

p4, L151: "However, in addition to given rise to MSC lines..." should be "...giving rise..."

p9, L376: "...The usefulness of this cell lines in bone and cartilage..." should be these cell lines.

The reviewer´s comments and the English grammar were revised throughout.

Reviewer 2 Report

The quality of the manuscript has been significantly improved. However, the following comments need to be addressed before the manuscript can be accepted for publication.

1. Line 223: "... one of them was incompletely immortalized." The authors should replace "one" with "two" because SCP-1 and BMA13H were incompletely immortalized.

2. There are a few typo errors at section 3.6. E.g., transplatation (line 353) and fibriogen (line 356). Line 353: the authors should remove the "," after "bone". Line 354: the authors should replace "this" with "these" and add "form" before "tumors".

Author Response

The quality of the manuscript has been significantly improved. However, the following comments need to be addressed before the manuscript can be accepted for publication.

We would like to thank you for your comments.

Line 223: "... one of them was incompletely immortalized." The authors should replace "one" with "two" because SCP-1 and BMA13H were incompletely immortalized.

This sentence has been changed as suggested.

There are a few typo errors at section 3.6. E.g., transplatation (line 353) and fibriogen (line 356). Line 353: the authors should remove the "," after "bone". Line 354: the authors should replace "this" with "these" and add "form" before "tumors".

These errors have been corrected.